# Microalga *Coelastrella* sp. Cultivation on Unhydrolyzed Molasses-Based Medium towards the Optimization of Conditions for Growth and Biomass Production under Mixotrophic Cultivation

**DOI:** 10.3390/molecules28083603

**Published:** 2023-04-20

**Authors:** Kamolwan Thepsuthammarat, Alissara Reungsang, Pensri Plangklang

**Affiliations:** 1Graduate School, Khon Kaen University, Khon Kaen 40002, Thailand; 2Department of Biotechnology, Faculty of Technology, Khon Kaen University, Khon Kaen 40002, Thailand; 3Research Group for Development of Microbial Hydrogen Production Process from Biomass, Khon Kaen University, Khon Kaen 40002, Thailand; 4Academy of Science, Royal Society of Thailand, Bangkok 10300, Thailand

**Keywords:** 3rd generation biomass, heterothrophy, photoautotrophy, amino acids, fatty acids

## Abstract

Improving biomass production with the utilization of low-cost substrate is a crucial approach to overcome the hindrance of high cost in developing large-scale microalgae production. The microalga *Coelastrella* sp. KKU-P1 was mixotrophically cultivated using unhydrolyzed molasses as a carbon source, with the key environmental conditions being varied in order to maximize biomass production. The batch cultivation in flasks achieved the highest biomass production of 3.81 g/L, under an initial pH 5.0, a substrate to inoculum ratio of 100:3, an initial total sugar concentration of 10 g/L, and a sodium nitrate concentration of 1.5 g/L with continuous light illumination at 23.7 W/m^2^. The photobioreactor cultivation results indicated that CO_2_ supplementation did not improve biomass production. An ambient concentration of CO_2_ was sufficient to promote the mixotrophic growth of the microalga as indicated by the highest biomass production of 4.28 g/L with 33.91% protein, 46.71% carbohydrate, and 15.10% lipid. The results of the biochemical composition analysis suggest that the microalgal biomass obtained is promising as a source of essential amino acids and pigments as well as saturated and monounsaturated fatty acids. This research highlights the potential for bioresource production via microalgal mixotrophic cultivation using untreated molasses as a low-cost raw material.

## 1. Introduction

Microalgae are photoautotrophic microorganisms capable of fixing CO_2_ and removing nutrients from wastewater. Their cultivation could lessen global climate change and water pollution problems. They are tolerant of environmental stress and have high growth rates. They are not seasonal or dependent on arable land [1,2]. *Coelastrella* spp. are green microalga which have been widely investigated for biotechnological application. *Coelastrella* spp. cells have the potential to produce carbohydrates, lipids and proteins [3]. Under stress cultivation conditions such as nutrient limitation, salt stress, low pH stress and high light exposure, the biomass of *Coelastrella* spp. comprised significant levels of lipids, carotenoids and antioxidants [4,5,6,7,8,9]. In addition, the production of mycosporine-like amino acids, a group of UV-protective compounds, was reported in *Coelastrella rubescens* under high light and UV-A exposition [7]. As *Coelastrella* spp. has the potential to produce and accumulate a broad range of valuable biocompounds, they have received considerable attention as they can be used as a feedstock for various value-added products such as food and feed supplements and biofuels as well as for biomedical, pharmaceutical, and nutraceutical products.

Microalgae are commonly grown under photoautotrophic conditions in which light is used as the sole energy source while inorganic carbon (mainly CO_2_) is supplied as a carbon source. Photoautotrophic cultivation usually yields low biomass concentrations because light penetration is one of the limiting factors in high-density culture [10]. Some microalgal strains can grow under heterotrophic conditions in which organic compounds are used as sources of energy and carbon under dark conditions. Mixotrophic cultivation is a combination of heterotrophic and photoautotrophic cultivation. Both inorganic and organic carbon sources can be consumed concurrently through photosynthetic and heterotrophic metabolism [11,12]. Therefore, this cultivation mode can mitigate limitations arising from microalgal biomass production under photoautotrophic culture [13]. However, the high cost of organic substrates makes this cultivation mode economically infeasible [14]. The optimization of process parameters for biomass production with the use of low-cost substrates might pave the way to overcoming this bottleneck.

Sugarcane molasses is a waste from the sugar industry which is rich in sugars, mainly sucrose, glucose, and fructose [15]. Therefore, it is considered as a potential feedstock to provide cheap organic carbon for microalgal growth. Previous research reported an efficient utilization of glucose and fructose for microalgal cultivation [16]. However, sucrose transportation systems are rarely detected in green microalgae. Many of them, such as *Chlorella* sp., Botryococcus sp., *Chlamydomonas* sp., and *Phormidium* sp. are unable to consume sucrose [17]. Waste molasses contains a significant amount of sucrose which is the main challenge in its direct utilization as a substrate for microalgae cultivation. Information on the utilization of unhydrolyzed molasses for green microalgae cultivation is limited to one report by Khanra et al. [18], who found that *Chlorococcum* sp. was able to grow and produced 18.88 g/L biomass under optimized mixotrophic cultivation conditions. Various published research studies reported the utilization of molasses for microalgae biomass production. However, pretreatment methods, such as enzymatic hydrolysis and chemical pretreatment, were applied to pretreat the molasses, yielding monosaccharides prior to the usage, which increased the overall substrate cost [19,20]. The present research discovered *Coelastrella* sp. KKU-P1 as the microalga capable of sucrose consumption and direct utilization of unhydrolyzed molasses. Therefore, it can be used for microalgae biomass production in a molasses-based medium without the extra cost of molasses saccharification.

Aside from the type of carbon source, the key factors affecting microalgal growth and biomass production are nutrient availability, substrate concentrations, light intensities, length of light/dark cycles, pH values, and cultivation temperature. Previous reports indicated that suitable carbon sources and mineral proportions could enhance microalgae growth and biomass production [21]. Excess nitrogen can promote the growth and protein production of microalgae, while starvation of biogenic elements such as nitrogen, phosphorus, and/or sulfur can induce carbohydrate or lipid accumulation but reduce the production of amino acid and protein, leading to slow microalgal growth [22]. Microalgal growth and biomass production depend on the pH of the culture medium. Microalgae are able to grow over a wide range of pH values, from 6.8 to 8.0. However, the suitable pH values vary depending on the microalgal species [13]. Light intensity and light–dark alteration generally affect photosynthetic activity, electron transfer, and ATP synthesis, which govern the electron flow toward the biomass and the production of storage materials [23,24]. An optimal light/dark cycle with suitable light intensity promotes efficient carbon assimilation through photosynthesis into biomass. Very high light intensities and long light periods can be the stress factors which promote the production of storage materials such as lipids and polyhydroxyalkanoates. However, these conditions can damage the chlorophyll, resulting in photo-inhibition and inefficient biomass production [18]. As various environmental conditions could affect microalgal cultivation, suitable levels of environmental factors should be investigated and controlled to achieve the highest microalgal growth and biomass production.

This research investigated the growth and biomass production of microalga *Coelastrella* sp. KKU-P1 in mixotrophic culture using molasses as a carbon source. Mixotrophic growth of the microalga on various sugars was evaluated compared to photoautotrophic growth. The effects of environmental conditions on microalgal growth and biomass production were investigated in flask-culture experiments. The obtained optimal conditions were further applied to evaluate the effect of CO_2_ concentrations in photobioreactor experiments. The biomass obtained under the optimal condition was analyzed for protein, carbohydrate, and lipid contents as well as amino acid and fatty acid profiles in order to evaluate the potential of biomass utilization.

## 2. Results and Discussion

### 2.1. Comparative Photoautotrophic and Mixotrophic Growth of Coelastrella *sp.* KKU-P1 with Different Sugars and Light Sources

The growth (biomass production) of *Coelastrella* sp. KKU-P1 in 3N Bold’s basal medium (3N BBM) with and without sugar supplementation is shown in Figure 1. A slight increase in the biomass concentration from 0.04 g/L to the range of 0.26–0.70 g/L was observed under photoautotrophic growth in BBM with no sugar (control). Visible light illumination of fluorescent and light-emitting diode (LED) lamps exhibited similar effects on microalga growth. Mixotrophic cultivation in BBM supplemented with sucrose, glucose, or fructose gave significantly higher biomass production (1–7 times) than that observed in the control. This suggested that the microalga could grow mixotrophically using sugars as carbon sources. Xylose or arabinose supplementation yielded lower biomass production. Lactose had no significant effect on microalga growth (Figure 1). At the end of incubation with a fluorescent light source, the highest biomass production was achieved in the treatments with fructose supplementation, followed by glucose and sucrose, in descending order (Figure 1a). In the treatments with an LED light source, maximal biomass production was also achieved by adding fructose and glucose to the medium. However, it was not markedly different from the values observed in the treatments with sucrose.

Adding sugars to the culture medium has been reported to enhance microalgae growth. Glucose and fructose can be converted to glyceraldehydes-3-phosphate, an important intermediate product involved in pentose phosphate and the Embden–Meyerhof–Parnas (EMP) pathway [25]. Glucose and fructose could enhance the mixotrophic growth of various microalgal strains such as *Scenedesmus* sp. [26], *Chorella* sp. [27], and *Monoraphidium* sp. [28].

Typically, disaccharides such as sucrose and lactose are hardly consumed by green microalgae. *Auxenochlorella protothecoides* could only grow in a medium with molasses after a pre-hydrolysis step, which confirmed its inability to utilize disaccharides [19]. In the present study, sucrose and its monomeric sugars (glucose and fructose) were good carbon sources for the mixotrophic growth of *Coelastrella* sp. KKU-P1. The results indicated that *Coelastrella* sp. KKU-P1 has the effective cellular activities to metabolize sucrose for its growth and metabolic functions. This would facilitate the development of microalga cultivation systems using waste materials, such as molasses, as low-cost carbon sources. The degradation of sucrose molecules into glucose and fructose by sucrose synthase or invertase enzyme has been reported in plant systems. Some green microalgae, such as *Tretadesmus dimorphus* [29], *Haemetococcus lacustris* [30], and *Chlorella vulgaris* [31], have shown the ability to consume sucrose, suggesting that their metabolism might involve sucrose degradation by this enzyme system. Lactose supplementation did not promote KKU-P1 growth. This implies that the microalga lacks the ability to produce β-galactosidase, a key enzyme for lactose hydrolysis. A similar growth pattern was observed in the treatment with lactose supplementation, and the control suggested that the microalga grew photoautotrophically in the presence of lactose. Girard et al. [32] reported similar results showing that lactose did not support the growth of *Chlorella vulgaris* and *A. protothecoides* under heterotrophic conditions.

Small amounts of pentoses (xylose or arabinose) were reported to enhance the growth of *Dunaliella salina* [33] and *Mychonastes homosphaera* [34]. However, an inhibitory effect of these sugars on the growth of *Coelastrella* sp. KKU-P1 was demonstrated in this study. The results implied that the strain KKU-P1 does not possess an inducible transport system for pentose. Additionally, microalga growth depending on energy derived from photosynthesis might be inhibited. Xylose or phosphorylated derivatives can slow microalgal growth by competing with xylulose 5-phosphate in the transketolase enzymatic system and blocking the photosynthetic carbon cycle [35].

Fructose supplementation yielded the highest biomass production when a fluorescent lamp was used as the light source, while relatively high biomass production could be achieved with fructose, glucose, and sucrose supplementation when an LED lamp was used (Figure 1). The photosynthetically active radiation (PAR) range covers wavelengths between 400–700 nm. LED and fluorescent lamps emit light that coincides with this wavelength range but with a unique emission spectrum. The color of light that finally reaches the microalgae cells depends on the light source spectrum, which directly affects microalgae growth and photosynthetic activities [36]. Since *Coelastrella* sp. KKU-P1 was able to grow mixotrophically under both light sources, the LED lamp was selected for the next experiments due to its long life and mercury-free characteristics [37].

### 2.2. Mixotrophic Cultivation of Coelastrella *sp.* KKU-P1 Using Molasses as a Carbon Source

#### 2.2.1. Effect of Initial pH

As shown in Figure 2a, the KKU-P1 strain could grow over a wide range of initial pH values, from 5.0 to 9.0. The microalga tended to grow with no lag period and to achieve high biomass production at initial pH values of 5.0 to 8.0. The medium with no pH adjustment had initial pH of 5.5 and exhibited a similar growth pattern to that observed with an initial pH of 5.0. A one-day lag period was observed with an initial pH of 9.0 and the microalga produced a lower biomass production compared to initial pH values of 5.0 to 8.0. An initial pH of 5.0 gave the maximal biomass concentration (1.43 g/L). Thus, it was considered the optimal initial pH for mixotrophic cultivation of the KKU-P1 strain and was applied as such in further experiments.

Figure 2b shows changes in pH values of the culture media over seven days of cultivation. Markedly increased pH values (1.20 to 1.40 fold) along with microalga growth was observed with initial pH values of 5.0 to 8.0. The photosynthetic activity exhibited during the mixotrophic growth of the microalga could generate carbonate and hydroxide ions, leading to an increased pH of the culture medium [38]. Additionally, nitrate consumption for microalgal biomass production under photoautotrophic cultivation also generates hydroxide ions that increase the medium’s pH [39]. Under an initial pH of 9.0, only a slight increase in the pH of the medium was observed with low biomass production. These results imply that under an initial pH of 9.0, the photoautotrophic growth of the KKU-P1 strain was inefficient.

#### 2.2.2. Effect of Substrate to Inoculum (S:I) Ratio and Initial Sugar Concentration

The effects of the S:I ratio on the biomass production and sugar consumption of *Coelastrella* sp. KKU-P1 at the end of cultivation are illustrated in Figure 3. The total sugar concentration was fixed at 5 g/L and the inoculum concentration was varied over the range of 0.025 to 0.20 g/L, which corresponded to a S:I ratio of 100:4 to 100:0.5 g/g. Increasing the inoculum concentration from 0.025 to 0.15 g/L significantly improved biomass production and sugar consumption. Further increasing the inoculum concentration to 0.2 g/L led to decreased biomass production and sugar consumption (Figure 3a). Therefore, a S:I ratio of 100:3 was considered a suitable condition for *Coelastrella* sp. KKU-P1 cultivation and was used in the next experiments. Higher S:I ratios than the optimal level typically inhibited microalgae growth and prolonged the lag phase as a result of substrate inhibition. A too low S:I ratio or an increase in the inoculum concentration to higher-than-optimal levels can cause a self-shedding effect and a shorter light exposure time for microalgae cells. With the application of a suitable S:I ratio, microalgae can adapt to the environment, reduce cell mortality, and increase growth and nutrient recovery, hence enhancing biomass production [40].

The effect of initial sugar concentrations on biomass production and sugar consumption at the end of *Coelastrella* sp. KKU-P1 cultivation was investigated. A suitable initial sugar concentration for biomass production was 10 g/L, yielding the highest biomass production of 2.9 g/L (Figure 3b). Higher or lower sugar concentrations than the optimal value significantly lowered biomass production (Figure 3b). Increasing the sugar concentration might increase the carbon to nitrogen and/or carbon to phosphate ratios to imbalance levels. Nitrogen- and phosphorus-limited conditions could retard microalgae growth while promoting the accumulation of energy-storage molecules such as starch and lipids [41]. Previous studies reported that growth of *M. reisseri* and *S. obliquus* was enhanced by sugarcane molasses supplementation to the culture medium at concentrations of 1 to 5 g/L [42,43]. In contrast, supplementation with 5 g/L of sugarcane molasses decreased the cell concentration of *Nostoc* sp. [44]. A high concentration of molasses caused a dark-colored medium and reduced light penetration, thus decreasing mixotrophic cultivation activity [45]. Additionally, harmful substances in molasses may have toxic effects on microalgal cells and reduce biomass production [46].

#### 2.2.3. Effect of Initial Nitrate Concentration

Variation of the initial sodium nitrate concentration over the range of 0.5 to 2.0 g/L yielded a biomass production at the end of cultivation of between 3.03 and 3.39 g/L. The maximal value was achieved with a sodium nitrate concentration of 1.5 g/L (Figure 4). Supplying high concentrations of sodium nitrate (1.5 and 2.0 g/L) enhanced the sugar consumption of *Coelastrella* sp. KKU-P1, while nitrate consumption was maximized at an initial sodium nitrate concentration of 1.5 g/L (Figure 4). Microalgae cells require sufficient nitrogen to support their growth under photoautotrophic, heterotrophic, and mixotrophic conditions. Sufficient nitrogen could enhance carbon fixation during photoautotrophic growth, thus increasing the overall biomass production [47]. Supplying high nitrate concentrations with balanced levels of carbon sources can promote microalgal growth [48]. However, nitrate concentrations which are higher than the optimal level could inhibit cell growth and metabolism. High nitrate concentrations were reported to enhance the activity of nitrate reductase, a key enzyme for converting nitrate to nitrite. An over-accumulation of nitrite in algal cells could inhibit their growth [49]. Additionally, more energy is needed by microalgae to convert nitrate into ammonium for cell assimilation, which could decrease the energy flux toward biomass production [50].

#### 2.2.4. Effect of Light Intensity and Light/Dark Cycle

*Coelastrella* sp. KKU-P1 was able to grow under heterotrophic cultivation (0 W/m^2^) using molasses as a carbon source and yielding a biomass concentration of 2.07 g/L at the end of cultivation (Figure 5a). The biomass production increased to 3.79 g/L when the light intensity was increased to 79 W/m^2^ (Figure 5a). Sugar consumption was not statistically different between any treatments; however, the lowest nitrate consumption, 56.47%, was observed under dark conditions. These results suggest that, under heterotrophic conditions, the microalga was able to grow and utilize sugars in the medium as carbon sources and nitrate as nitrogen sources. Additionally, light illumination facilitates photoautotrophic growth in which CO_2_ (naturally present in the air and/or resulting from sugar degradation) was consumed as a carbon source while nitrate was used as a nitrogen source. Therefore, mixotrophic cultivation could promote biomass production of the strain KKU-P1 as compared to cultivation under heterotrophic (Figure 5a) and photoautotrophic conditions (Figure 1).

The effects of the light/dark cycle on biomass production were illustrated in Figure 5a. The results indicate that increased photoperiods promoted microalgal growth. The highest biomass production, 3.81 g/L, was obtained under a continuous light illumination (24:0 light/dark cycle) and was 1.5 and 1.8 fold higher than 12:12 and 0:24 light/dark cycles, respectively. An insignificant difference in sugar consumption was observed in all experiments, while lower nitrate consumption was noticed under dark conditions (0:24 light/dark cycle) compared to other treatments (Figure 5a). These results suggest that a 24:0 light/dark cycle was optimal for the biomass production of *Coelastrella* sp. KKU-P1. A shorter light period may act as a stress factor, causing a lower rate of cell division [51]. A similar effect of the light/dark cycle on *A. protothecoides* growth was reported by Patel et al. [52] in which continuous light illumination yielded a 2.07 fold higher biomass concentration than a 16:8 light/dark cycle at the same light intensity (22.5 W/m^2^).

### 2.3. Mixotrophic Cultivation of Coelastrella *sp.* KKU-P1 Using Molasses and CO_2_ as Carbon Sources

CO_2_ concentration affected the biomass production of *Coelastrella* sp. KKU-P1. The negative effect of CO_2_ supplementation on the microalgal biomass production is shown in Figure 6a in which it can be seen that biomass production decreased with increased CO_2_ concentration. The highest biomass production (4.28 g/L) with the maximum sugar consumption (88.11%) and maximum nitrate consumption (98.66%) was achieved by supplementing the air with an ambient CO_2_ concentration (approximately 0.03%) during microalga cultivation.

The biochemical compositions of the biomass obtained at the end of cultivation were examined to illustrate the potential of biomass utilization and the results are depicted in Figure 6b. Supplying air with an ambient CO_2_ concentration yielded biomass with a protein content of 33.91%, which was not statistically different from that obtained in treatments with 2% or 10% CO_2_ aeration. The carbohydrate content obtained in the treatment with ambient CO_2_ concentration was similar to that observed with 2% CO_2_; however, a markedly lower carbohydrate content was observed with 10% CO_2_. The significant decrease in lipid contents as the result of CO_2_ supplementation is also demonstrated in Figure 6b.

Previous studies reported that the photoautotrophic cultivation of *Coelastrella* spp. with 1.5–5.0% CO_2_-supplemented air yielded 1.2 to 4 times higher biomass concentrations [53,54]. *Coelastrella* sp. FI69 and *Coelastrella rubescens* accumulated higher lipid when CO_2_ was supplemented during photoautotrophic cultivation [8,54]. CO_2_ supplementation under mixotrophic conditions could enhance photosynthesis. However, excess CO_2_ might obstruct organic carbon metabolism, thus decreasing the overall efficiency of mixotrophic microalgal cultivation [55]. A high concentration of CO_2_ can cause the pH of the medium to decrease due to the conversion of excess CO_2_ into carbonic acid that accumulates in the culture medium [56]. This decreases the activity of RuBisCO, a key enzyme in photosynthesis [57], and therefore inhibits microalgal growth and metabolic activities. An inhibitory effect of high CO_2_ concentrations on microalgal metabolism has been reported for various microalgae such as *H. lacustris* [58], *Chlorella vulgaris* [59], *Parachlorella kessleri* [60], and *Coelastrella* sp. [54].

Overall, the results suggest that CO_2_ at an ambient concentration was sufficient to promote the growth, and the carbohydrate and lipid accumulation, of *Coelastrella* sp. KKU-P1 under mixotrophic cultivation. This might have been because microalgae have carbon-dioxide-concentrating mechanisms (CCMs) to acquire and concentrate inorganic carbon from the culture medium. The CCMs are induced under low CO_2_ conditions, resulting in efficient carbon dioxide utilization during microalgal photosynthesis [47].

The biomass of *Coelastrella* sp. KKU-P1 cultivated in the photobioreactor under the optimal conditions was analyzed for amino acid and pigment contents, as well as to determine its fatty acid profile. Microalga biomass contains both essential and non-essential amino acids (Table 1). Alanine, glutamic acid, glycine, and aspartatic acid were the primary non-essential amino acids detected in the biomass. All essential amino acids, except tryptophan, were found in the biomass. It is noted that asparagine and selenocysteine were not analyzed in this research. Tryptophan was previously detected in the biomass of *Coelastrella terrestris* and *Coelastrella* sp. LRF1 [61,62] and increased tryptophan contents were observed in UV-A- and UV-B-irradiated cells [61]. This amino acid was undetectable in *Coelastrella* sp. KKU-P1 biomass, which might be due to the different cultivation conditions applied. In addition, in this study, the biomass was hydrolyzed by 6 M HCl at 110 °C for 22 h prior to the analysis, which might have caused the destruction of tryptophan [63]. Leucine and valine were found in the highest concentrations. These two amino acids were reported to promote protein synthesis in fish as well as to improve meat quality and immune responses in other monogastric species [64]. Supplying sea urchins with alanine or glutamic acid supplemented feed resulted in a desirable sweet taste for sea urchin gonads [65]. Glycine plays an important role in protein synthesis for mammals and fish. However, previous studies showed that the glycine synthesized in-vivo was insufficient to meet metabolic demands [66]. An L-aspartic acid pretreatment of fish is a preventive strategy against *Vibrio alginolyticus* infection [67]. Lysine and methionine are limiting amino acids. Many microalgal strains are comprised of high levels of lysine but some species are deficient in cysteine and methionine, the sulfur-containing amino acids. The amino acid profiles indicate that the biomass of *Coelastrella* sp. KKU-P1 might be a promising amino acid source for animal feed. Additionally, microalga biomass contains 9.90 ± 0.01 mg/g chlorophyll a, 6.31 ± 0.45 mg/g chlorophyll b, and 1.39 ± 0.16 mg/g carotenoid which could be supplemented into feed to improve animal meat and egg quality. *Coelastrella* sp. LRF1 accumulated lower contents of chlorophyll a and b (5.22 and 1.95 mg/g, respectively) with a slightly higher content of carotenoid 1.69 mg/g [62]. Carotenoid contents of higher than 2.0 mg/g were observed when *Coelastrella* spp. were cultivated under stress conditions such as high light intensity and osmotic stress [3].

The fatty acid profile of the lipid extracted from *Coelastrella* sp. KKU-P1 biomass is given in Figure 7. These extracted lipids are rich in monounsaturated and unsaturated fatty acids, accounting for 49.70% and 43.68% of the lipid fraction, respectively. The most common unsaturated fatty acid is stearic acid (18:0) (40.78%), while the major monounsaturated fatty acids are vaccenic acid (18:1n7) (34.73%) and palmitoleic acid (16:1) (11.16%). Low contents of polyunsaturated fatty acids (PUFA) (2 or 3 double bonds) (6.62%) were detected and no polyunsaturated acids with ≥4 double bonds were found. The results revealed that lipid extracted from *Coelastrella* sp. KKU-P1 biomass is a favorable feedstock for biodiesel production. *Coelastrella* spp. were reported to accumulate high contents of polyunsaturated fatty acids (45.6–65.2%) under photoautotrophic cultivation [8,53,68,69]. In contrast, the mixotrophic cultivation of *Coelastrella* spp. yielded biomass containing mainly monounsaturated and unsaturated fatty acids (67.8–91.2%) [69,70,71], which coincided with the results obtained in the present study.

## 3. Materials and Methods

### 3.1. Seed Microalga Preparation

The freshwater microalga, *Coelastrella* sp. KKU-P1 (accession number MW581273) isolated from a fish pond in Nakhon Ratchasima Province in Thailand, was used for microalgal biomass production in this study. A seed inoculum was prepared by cultivating the strain KKU-P1 in a 500 mL Erlenmeyer flask containing 250 mL of 3N BBM [72] supplemented with glucose (5 g/L) as a carbon source. The flask was incubated at 30 °C with continuous illumination at 23.7 W/m^2^ in an orbital shaker at 150 rpm for 14 days. Microalgal cells were harvested by centrifugation at 4000 rpm for 5 min and washed two times with distilled water before being used as a seed inoculum in further experiments.

### 3.2. Effects of Sugar Supplementation and Light Source on Mixotrophic Growth of Coelastrella *sp.* KKU-P1

The growth of *Coelastrella* sp. KKU-P1 in 3N BBM supplemented with various kinds of sugars, including glucose, fructose, sucrose, lactose, arabinose, and xylose, at a concentration of 5 g/L, corresponding to 28, 28, 15, 15, 33, and 33 mM, respectively, was investigated. A microalga growth assay with 3 replications was conducted in 96-well microplates containing 200 μL of culture media and 50 μL of seed inoculum. The initial cell concentration was approximately 10^4^ cells/mL, corresponding to an OD_655_ value of 0.005 and a biomass concentration of 0.11 g/L. The microplates were incubated at 30 °C and were continuously illuminated on the top of the plate by visible light from a cool white fluorescent lamp or a cool white LED lamp at approximately 23.7 W/m^2^ for 10 days. The cultures were kept under axenic conditions throughout the experiment. The OD_655_ values of the culture were measured every 24 h using a microplate reader (iMark, Bio-Rad, Hercules, CA, USA). The microplates were shaken at a medium speed, set on an instrument function, for 30 s prior to the measurement. The OD_655_ values were converted to the biomass concentration using the calibration curve in which one unit of OD_655_ is equal to a biomass concentration of 5.84 g/L (R^2^ = 0.998).

### 3.3. Mixotrophic Cultivation of Coelastrella *sp.* KKU-P1 Using Molasses as a Carbon Source

Mixotrophic cultivation of *Coelastrella* sp. KKU-P1 was carried out in 500 mL Erlenmeyer flasks containing 250 mL of BBM supplemented with unhydrolyzed molasses as the sole carbon source. The molasses comprised 60.43 ± 2.28% total sugar, 0.52 ± 0.12% total nitrogen, 72.11 ± 3.94% total solids, and 10.59 + 1.32% ash of dry matter. The results closely agreed with the molasses compositions reported elsewhere [73,74,75]. The effects of environmental factors, including initial pH, initial total sugar (TS) concentration, initial NaNO_3_ concentration, light intensity, and length of the light/dark cycle were investigated using a one-factor-at-a-time design. Unless otherwise stated, microalga cultivation with three biological replicates was conducted under a TS concentration of 5 g/L, an initial biomass concentration of 0.1 g/L, a NaNO_3_ concentration of 0.75 g/L, and with incubation at 30 °C with a shaking speed of 150 rpm and continuous illumination at approximately 23.7 W/m^2^. The cultivation was carried out until the stationary phase of growth was reached.

The effects of initial pH on microalgal biomass production were investigated first by varying the culture pH from 5.0 to 9.0. The initial pH of the medium was adjusted by 1.0 M HCl or NaOH solution. Culture samples were taken every 24 h for the analysis of biomass concentration, total sugar concentration, and pH. The initial pH that yielded the highest biomass was used in further experiments. The influences of the substrate to inoculum (S:I) ratio were investigated at the initial TS concentration of 5 g/L by varying the ratios of TS concentration (g/L) to inoculum concentration (g/L) as 100:4, 100:3, 100:2, 100:1, and 100:0.5. The biomass and sugar concentrations were analyzed every 24 h to assess the optimal S:I ratio for biomass production, which was then applied to investigate the effects of initial sugar concentrations 5, 10, 15, 20, 25, and 30 g/L.

Under the optimal initial pH, S:I ratio, and the initial sugar concentration, the NaNO_3_ concentration in the culture media was varied as 0.5, 0.75, 1.0, 1.5, and 2.0 g/L (corresponding to nitrate concentrations of 0.37, 0.57, 0.73, 1.09, and 1.46 g/L, respectively) to examine the effect of initial nitrate concentration on microalgal biomass production. Culture samples were taken every 24 h to analyze biomass, TS, and nitrate concentrations. The optimal initial sodium nitrate concentration was then used to examine the effects of light intensities (23.7 and 79 W/m^2^), and the obtained optimal conditions were then applied in an investigation of the effects of the light/dark cycle (24/0, 18/6, 12/12, and 0/24 h/h) on biomass production. The optimal conditions obtained in the flask experiments were used in the mixotrophic cultivation of *Coelastrella* sp. KKU-P1 with supplementation of molasses and CO_2_ as carbon sources in a photobioreactor (PBR).

### 3.4. Mixotrophic Cultivation of Coelastrella *sp.* KKU-P1 Using Molasses and CO_2_ as Carbon Sources

The microalga was cultivated in a one-liter batch PBR containing 800 mL of BBM supplemented with molasses and NaNO_3_ at the optimal levels obtained in the previous experiments. Experiments were conducted under the optimal initial pH, F:I ratio, light intensity, and light/dark cycle length. Microalga cultivation was carried out in a temperature-controlled room at 25 ± 3 °C with continuous illumination of one side of the reactor (23.7 W/m^2^, measured on the PBR surface). The PBR was continuously mixed using a magnetic stirrer. Air supplemented with CO_2_ at various concentrations of 0, 2, and 10% (*v*/*v*) was supplied to the PBR at a rate of 0.2 vvm. Cultivation was conducted until the stationary phase of growth was reached. All the experiments were conducted in biological triplicates.

### 3.5. Analytical Methods

The culture samples were centrifuged at 10,000 rpm for 5 min. The culture supernatant was used for the analysis of total sugar and nitrate concentrations. Total sugar concentration was analyzed using the phenol sulfuric method described by Dubois et al. [76]. Nitrate concentration was measured according to the APHA method [77].

Cell pellets were washed using distilled water, re-centrifuged two times, and further used for the determination of biomass concentration and biomass composition (carbohydrate, lipid, and protein contents). The resulting microalgal biomass was dried at 80 °C to a constant weight to determine the cell dry weight, expressed as the microalgal biomass concentration in grams of biomass per liter of culture (g/L). The total carbohydrate and protein contents (% of cell dry weight) of the biomass were analyzed according to Pruvost et al. [78]. Lipid content (% of cell dry weight) was analyzed following the methods of Mishra et al. [79].

Biomass was disrupted using osmotic shock following the methods of Byreddy et al. [80] before lipid extraction following Huang et al. [81]. The fatty acid compositions of the extracted lipids were analyzed using gas chromatography, according to Huang et al. [81]. Total chlorophyll and total carotenoid determinations were performed as described by Singh et al. [82]. The amino acid profile of the biomass was analyzed using an Amino Acid Analyzer (Hitachi, L-8900, Tokyo, Japan).

### 3.6. Statistical Analysis

All flask and PBR experiments were performed in biological triplicates. Mean values ± SD are presented in the figures. The effects of variable factors on biomass production and composition were determined with analysis of variance (ANOVA) followed by Duncan’s multiple-ranged test using the SPSS program Version 17.0 (SPSS Inc., Chicago, IL, USA). A significant difference between treatments was identified as those having *p*-values less than or equal to 0.05.

## 4. Conclusions

This research demonstrates the potential of using unhydrolyzed molasses as a low-cost carbon source for the mixotrophic cultivation of microalga *Coelastrella* sp. KKU-P1. An initial pH of 5.0, a substrate to inoculum ratio of 100:3, and sodium nitrate and sugar concentrations of 1.5 g/L and 10 g/L, respectively, are optimal for microalgal biomass production. High light intensity with continuous illumination promoted biomass production. CO_2_ at an ambient concentration was sufficient to promote microalgal biomass production as well as carbohydrate and lipid accumulation under mixotrophic cultivation. Overall, a high biomass production of 4.28 g/L with 33.91% protein, 46.71% carbohydrate, and 15.10% lipid could be achieved in the mixotrophic cultivation of *Coelastrella* sp. KKU-P1 under the obtained optimal conditions. Biomass with a high carbohydrate content can be used as a raw material to produce biofuels such as bioethanol and biogas. The fatty acid profile of lipids extracted from microalgal biomass is desirable for biodiesel production. Additionally, biochemical composition analysis suggested that the obtained microalgal biomass is a promising source of essential amino acids and pigments that can potentially be used as an animal feed supplement.

## Figures and Tables

**Figure 1 molecules-28-03603-f001:**
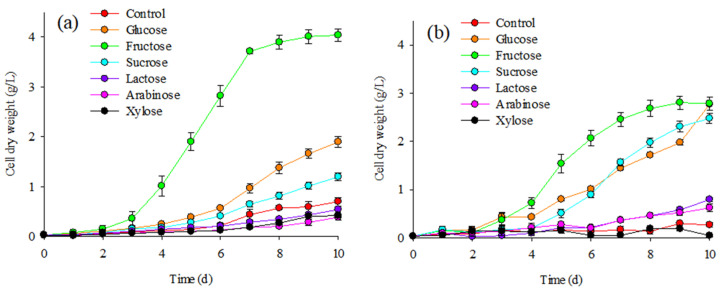
Photoautotrophic and mixotrophic growth of *Coelastrella* sp. KKU-P1 with various carbon sources under visible light illumination from (**a**) fluorescent or (**b**) light-emitting diode lamps. Average values of cell concentration expressed as cell dry weight were presented along with the standard deviation values of three replicates.

**Figure 2 molecules-28-03603-f002:**
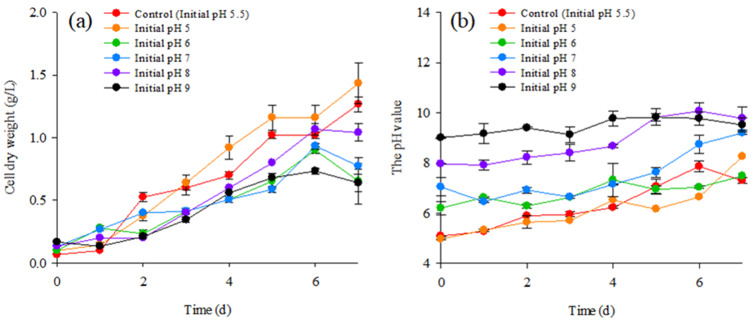
Effect of initial pH on (**a**) biomass production and (**b**) medium pH during mixotrophic cultivation of *Coelastrella* sp. KKU-P1 in molasses-based medium.

**Figure 3 molecules-28-03603-f003:**
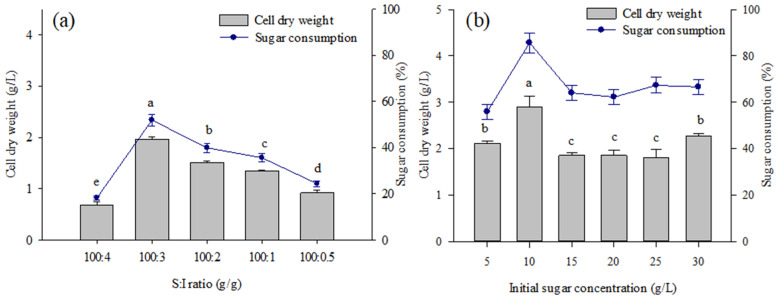
Effect of (**a**) substrate to inoculum (S:I) ratio and (**b**) initial sugar concentration on biomass production and sugar consumption of *Coelastrella* sp. KKU-P1 cultivated in a molasses-based medium. Different lowercase letters on the top of the bars indicate significant effects of the S:I ratio or initial sugar concentrations on biomass production (*p* ≤ 0.05, ANOVA with Duncan’s multiple range test).

**Figure 4 molecules-28-03603-f004:**
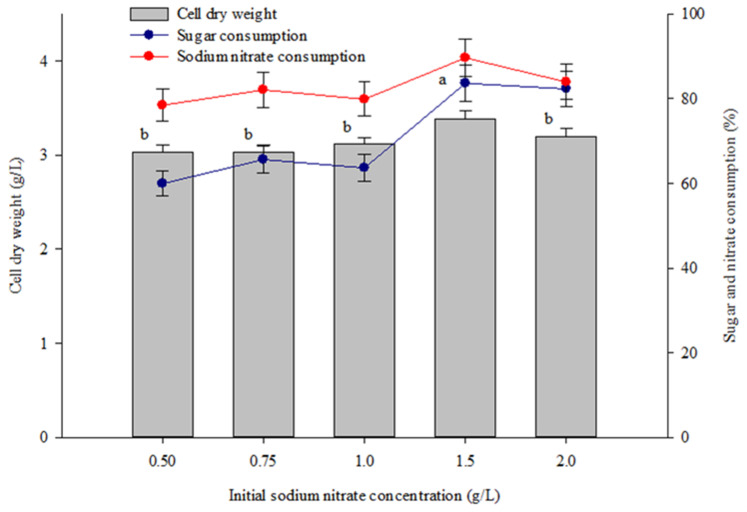
Effect of initial sodium nitrate concentration on biomass production and percentage of sugar and nitrate consumption of *Coelastrella* sp. KKU-P1 cultivated in molasses-based medium. Different lowercase letters on the top of the bars indicate the significant effects of the initial sodium nitrate concentrations on biomass production (*p* ≤ 0.05, ANOVA with Duncan’s multiple range test).

**Figure 5 molecules-28-03603-f005:**
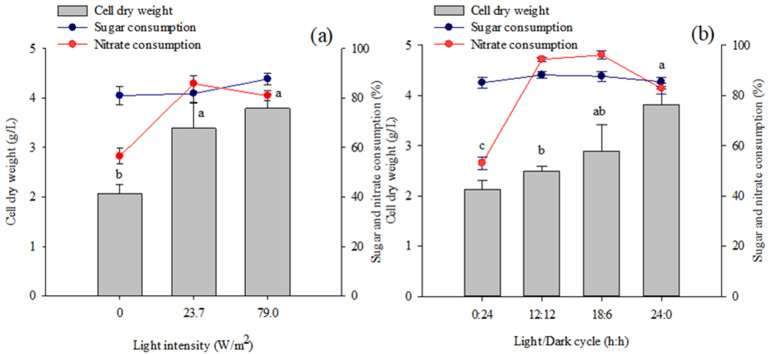
Effect of (**a**) light intensity and (**b**) light/dark cycle length on biomass production and percentage of sugar and nitrate consumption of *Coelastrella* sp. KKU-P1 cultivated in molasses-based medium. Different lowercase letters on the top of the bars indicate significant effects of light intensity or length of light/dark cycles on biomass production (*p* ≤ 0.05, ANOVA with Duncan’s multiple range test).

**Figure 6 molecules-28-03603-f006:**
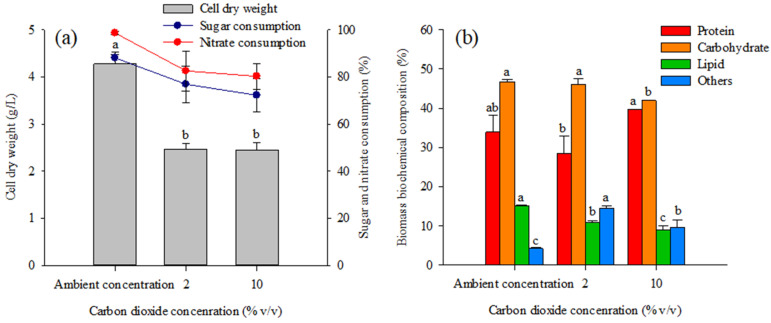
Effect of CO_2_ concentration on (**a**) biomass production and sugar and nitrate consumption and (**b**) composition of *Coelastrella* sp. KKU-P1 biomass obtained from cultivation in molasses-based medium. Different lowercase letters on the top of the bars indicated that the data are significantly different (*p* ≤ 0.05, ANOVA with Duncan’s multiple range test).

**Figure 7 molecules-28-03603-f007:**
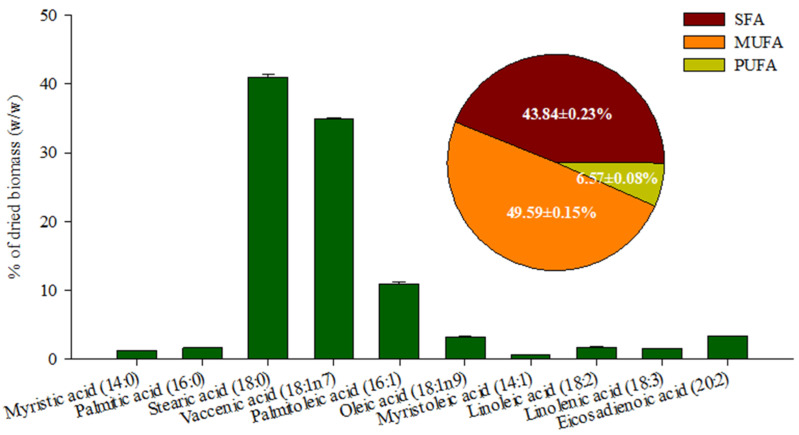
Contents of saturated fatty acids (SFA), monounsaturated fatty acids (MUFA), polyunsaturated fatty acids (PUFA) and fatty acid profiles of lipid extracted from *Coelastrella* sp. KKU-P1 biomass.

**Table 1 molecules-28-03603-t001:** Amino acid profile of *Coelastrella* sp. KKU-P1 biomass.

Essential Amino Acids (EAA)	nmol/mg	Non-Essential Amino Acids (NEAA)	nmol/mg
Threonine	68.74 ± 9.17	Aspartic acid	126.96 ± 16.94
Valine	109.64 ± 14.63	Serine	75.94 ± 10.13
Methionine	26.38 ± 3.52	Glutamic acid	157.16 ± 20.97
Isoleucine	50.50 ± 6.74	Proline	53.30 ± 7.11
Leucine	126.00 ± 16.81	Glycine	150.44 ± 20.08
Phenylalanine	60.92 ± 8.13	Alanine	196.86 ± 26.27
Histidine	24.40 ± 3.26	Cysteine	3.18 ± 0.42
Lysine	62.20 ± 8.30	Tyrosine	26.50 ± 3.54
Arginine	64.90 ± 8.66		
Tryptophan	Not detected		

## Data Availability

The datasets used and/or analyzed during the current study are available from the corresponding author on reasonable request.

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
