# Peer review of "Microalga Coelastrella sp. Cultivation on Unhydrolyzed Molasses-Based Medium towards the Optimization of Conditions for Growth and Biomass Production under Mixotrophic Cultivation"

_molecules, 2023, doi:10.3390/molecules28083603_

Round 1

Reviewer 1 Report

The authors investigated molasses as carbon source for microalgal growth. This topic is highly interesting also in an industrial point of view since the use of side streams are important to lower production costs. The article is well written, and the results are adequately compared to the literature. 

Nevertheless, following changes should be considered:

Figure 1. Mention growth in 96-well plates in Figure caption. What was the n? Add statistics. Convert OD to biomass concentration or cell count. OD could also increase due to the growth of bacteria. Was the culture kept axenic through all the trials?

Line 67: reference on assimilation of sucrose by Coelastrella sp. KKU-P1

Line 131: delete “the”

Line 205: delete “a”

Line 210: delete “the” from microalgae

Line 298: delete “the”

Which type of LED was used, cold white or warm white (K)?

What was used to adjust the pH?

Statistical test for microplate growth test is missing. Were the plates agitated?

Statistics of amino acid and fatty acid profile is missing. What was the n?

Author Response

Reviewer #1

  1. Figure 1. Mention growth in 96-well plates in Figure caption. What was the n? Add statistics. Convert OD to biomass concentration or cell count. OD could also increase due to the growth of bacteria. Was the culture kept axenic through all the trials?

Response: We converted the OD to biomass concentration, as shown in Figure 1. The microplate experiment was conducted in triplicates (page 13 line 413). Average values of cell concentration expressed as cell dry weight were presented in the graph along with the standard deviation values of three replicates (Figure 1).  We also added this statement to the caption of Figure 1. In addition, the plates were kept axenic throughout the experiment. We added this information in page 13 lines 418-419.

  1. Line 67: reference on assimilation of sucrose by Coelastrella sp. KKU-P1

Response: The statement about the sucrose assimilation by Coelastrella sp. KKU-P1 is from our study results. To avoid confusing, we revised the sentence to “The present research discovered Coelastrella sp. KKU-P1 as the microalga capable of sucrose assimilation and direct utilization of unhydrolyzed molasses. Therefore, it can be used for microalgae biomass production in a molasses-based medium without the extra cost of molasses saccharification.” (page 2 lines 72-76).

  1. Line 131: delete “the”

Response: We revised as recommended (page 4 line139).

  1. Line 205: delete “a”

Response: We revised as recommended (page 6 line 215).

  1. Line 210: delete “the” from microalgae

Response: We revised as recommended (page 6 line 217).

  1. Line 298: delete “the”

Response: We revised as recommended (page 10 line 309).

  1. Which type of LED was used, cold white or warm white (K)?

Response: The cool white LED was used in this experiment. We added this information in page 13 lines 417-418.

  1. What was used to adjust the pH?

Response: NaOH (1.0 M) and HCl (1.0 M) were used to adjust pH of the culture media. We added the detail of pH adjustment as recommended (page 14 lines 439-440).

  1. Statistical test for microplate growth test is missing. Were the plates agitated?

Response: Since the microalga growth was presented as a line graph, we did not add the statistical test to the graph. However, the standard deviation from 3 replicate values was added along with the average values (Figure 1).

  1. Statistics of amino acid and fatty acid profile is missing. What was the n?

Response: The data of fatty acid and amino acid profiles were obtained from 3 replicate experiments.  We put the standard deviation along with average values as shown in Table 1 and Figure 7.

Reviewer 2 Report

REVIEW OF THE ARTICLE BY KAMOLWAN THEPSUTHAMMARAT ET AL. ENTITLED “MICROALGA COELASTRELLA SP. CULTIVATION ON UNHYDROLYZED MOLASSES-BASED MEDIUM TOWARDS CONDITIONS OPTIMIZATION FOR GROWTH AND BIOMASS PRODUCTION UNDER MIXOTROPHIC CULTIVATION” (molecules-2306771)

The authors studied growth parameters of the strain of green microalga Coelastrella sp. KKU-P1 cultivated mixotrophically in photobioreactors. They also studied its chemical composition and proposed the strain as a prospective object for biotechnology. It is a routine study, but bringing some new useful data on the effect of molasses and different sugars on algal growth. The article is in scope of the journal. It should be revised according to the specific comments (please, see below).

l. 17. It is better to replace 5.0:0.15 with 100:3 g/g to avoid decimal separators in proportions.

l. 18-19. ‘Biophotoreactor’ should be ‘photobioreactor’.

l. 28-99. In the Introduction I suggest focusing on the biotechnological significance of Coelastrella spp. as producers of carotenoids, fatty acids, mycosporine-like amino acids and other valuable compounds as well as its possible biotechnological applications (see doi.org/10.3390/md20030175, doi.org/10.3390/plants10122601, 10.3390/md21020108, doi.org/10.1016/j.fuel.2020.118490).

l. 33-34. The sentence should be revised. Polysaccharides are also carbohydrates, fatty acids are also lipids, main algal pigments (carotenoids and chlorophylls) are also lipids.

l. 38-45. References are missing.

l. 41. It is wrong. Main limiting factor is air CO2 concentration.

l. 43. “Mixotrophic cultivation is a variant of heterotrophic culture.” - the statement is unclear. Cultivation is not a culture. There are three strategies of cultivation: phototrophic, mixotrophic and heterotrophic. Mixotrophic is not a type of heterotrophic.

l. 45, 131. The term assimilation refers to the inorganic carbon only.

l. 45. Please, revise: respiratory metabolism can be also ‘phototrophic’, or light-mediated. Do you mean phototrophic and heterotrophic?

l. 66. Reference is required.

l. 70-74. References are missing.

l. 75. Do you mean biogenic element starvation? Phosphorus and nitrogen are not minerals used for algal culturing, but elements.

l. 108-111. The conclusion is wrong. Mixotrophic cultivation can show the ability to mixotrophy (not heterotrophy).

l. 107, 113, 128, 157, 161, 382. Do you mean luminescent light source?

l. 121. Pentose phosphate pathway is also glycolysis. Do you mean pentose phosphate and Embden-Meyerhof-Parnas pathway?

l. 132, 148, 280, Correct name of C. protothecoides is Auxenochlorella protothecoides

l. 140. Correct name of Scenedesmus dimorphus is Tetradesmus dimorphus.

l. 141, 148, 150, 310, To avoid confusion, please, use the full name for Chlorella and Coelastrella. Because C. could refer to Chlorella, Coelastrella and Chlorococcum.

l. 144. Beta should be β.

l. 150. Correct name of Chlorella minutissima is Mychonastes homosphaera.

l. 150. D. salina should be written in full.

l. 150. It was only in the case of luminescent lamps. How can it be explained?

l. 155. “phosphorus uptake in photosynthesis” - what is it? Photosynthesis is the process in which a phototrophic cell uses sunlight to synthesize nutrients from carbon dioxide and water.

l. 161-162. For this statement emission spectra should be given as supplementary materials.

Which sugar was selected for further studies as a result of the subsection 2.1? It should be indicated in the end.

l. 177. ‘it is considered’ should be ‘it was considered’.

l. 195-296. It is not clear, how do you define growth/productivity of the culture. For example, the phrases like “Further increasing inoculum concentration to 0.2 g/L led to the reduction of biomass concentration” are confusing. In the case inoculum concentration of 0.2 g/L the biomass concentration is also 0.2 g/L, because you inoculated the biomass. 

l. 196-175. What do you mean on feed here?

 Figure 3a. What are the units of F:I? g/g? Please indicate in the figure and at the first mention in the text.

l. 297-310, Figure 7. Please, indicate, it was the volumetric CO2 percentage (v/v).

Figure 3,4,5,6. Please, indicate in the legend, how is ‘growth’ defined? As culture dry mass at the female stage of the experiment?

l. 239. What about mixotrophic conditions?

l. 272, 395. It is not cell concentration, but biomass concentration.

l. 250-282, 413.  Figure 5. lux are units of illuminance, not light intensity. Units of light intensity are W/m2.

Table 1. There are some questions about the amino acid profile. First, what is about asparagine, glutamine and selenocysteine? Second, how can you explain the absence of free tryptophan? Coelastrella has many W-containing proteins.

l. 291-296. Please, compare you results on CO2 effect with other works on Coelastrella spp. (doi.org/10.1134/S1021443716040105, doi.org/10.1016/j.bcab.2018.03.022).

l. 310. Correct name of Chlorella kesslerii is Parachlorella kesslerii. Correct name of Haematococcus pluvialis is Haematococcus lacustris.

l. 342. Please, compare with other data on Coelastrella.

l. 344-352. Please, discuss fatty acid composition compared to other studied Coelastrella spp. (doi.org/10.3390/life12040560, doi.org/10.1134/S1021443716040105, doi.org/10.4490/algae.2017.32.8.6, doi.org/10.1016/j.bcab.2018.03.022, doi.org/10.3390/life12030334).

l. 364-383. The weakness of the methodology is using mass concentrations of sugars instead of molar ones. Obviously, the same mass of pentoses and hexoses contains different  amounts of sugar molecules. It could affect growth. This limitation should be mentioned; I suggest recalculate molar concentrations.

l. 379. μl  should be μL.

l. 387-389. Have you studied or have you found published data on composition of used molasse? It would be nice to discuss it in Methods and/or Discussion.

l. 427. %-v/v?

Author Response

Reviewer #2

  1. 17. It is better to replace 5.0:0.15 with 100:3 g/g to avoid decimal separators in proportions.

Response: We revised the values of the substrate to inoculum (S:I) ratio by setting the substrate (S) value to 100 as recommended throughout the manuscript.

  1. 18-19. ‘Biophotoreactor’ should be ‘photobioreactor’.

Response: We revised as recommended (page 1 line 19).

  1. 28-99. In the Introduction I suggest focusing on the biotechnological significance of Coelastrella spp. as producers of carotenoids, fatty acids, mycosporine-like amino acids and other valuable compounds as well as its possible biotechnological applications (see doi.org/10.3390/md20030175, doi.org/10.3390/plants10122601, 10.3390/md21020108, doi.org/10.1016/j.fuel.2020.118490).

Response: We revised the introduction with focusing on the biotechnological significance of Coelastrella sp. as recommended (pages 1-2 lines 32-43).

  1. 33-34. The sentence should be revised. Polysaccharides are also carbohydrates, fatty acids are also lipids, main algal pigments (carotenoids and chlorophylls) are also lipids.

Response: Since the introduction was revised with focusing on the biotechnological significance of Coelastrella sp. (pages 1-2 line 32-43), this statement was revised accordingly.

  1. 38-45. References are missing.

Response: References for this statement were added to the text (page 2 line 47)  and reference list (page 2 line 52).

  1. 41. It is wrong. Main limiting factor is air CO2 concentration.

Response: We revised the sentence to “Photoautotrophic cultivation usually yields low biomass concentrations because light penetration is one of the limiting factors in high-density culture (page 2 lines 46-47).

  1. 43. “Mixotrophic cultivation is a variant of heterotrophic culture.” - the statement is unclear. Cultivation is not a culture. There are three strategies of cultivation: phototrophic, mixotrophic and heterotrophic. Mixotrophic is not a type of heterotrophic.

Response: We revised the sentence to “Mixotrophic cultivation is a combination of heterotrophic and photoautotrophic cultivation.  Both inorganic and organic carbon sources can be consumed concurrently through photosynthetic and heterotrophic metabolism” (page 2 lines 49-52).

  1. 45, 131. The term assimilation refers to the inorganic carbon only.

Response: We changed the term “assimilation” to “consumed” throughout the manuscript when the organic carbon (sugar) was referred.

  1. 45. Please, revise: respiratory metabolism can be also ‘phototrophic’, or light-mediated. Do you mean phototrophic and heterotrophic?

Response: We changed “respiratory metabolism” to “heterotrophic” as shown in  (page 2 line 52)

  1. 66. Reference is required.

Response: The statement about the sucrose assimilation by Coelastrella sp. KKU-P1 is from our study results. To avoid confusing, we revised the sentence to “The present research discovered Coelastrella sp. KKU-P1 as the microalga capable of sucrose assimilation and direct utilization of unhydrolyzed molasses. Therefore, it can be used for microalgae biomass production in a molasses-based medium without the extra cost of molasses saccharification.” (page 2 lines 72-76).

  1. 70-74. References are missing.

Response: Reference for this statement was added to the text (page 2 line 81)  and reference list (page 17 line 586-587).

  1. 75. Do you mean biogenic element starvation? Phosphorus and nitrogen are not minerals used for algal culturing, but elements.

Response: We changed “starvation of minerals” to “starvation of biogenic elements” as recommended (page 2 line 82)

  1. 108-111. The conclusion is wrong. Mixotrophic cultivation can show the ability to mixotrophy (not heterotrophy).

Response: We revised the statement to “This suggested that the microalga could grow mixotrophically using sugars as carbon sources.” (page 3 lines 118-119).

  1. 107, 113, 128, 157, 161, 382. Do you mean luminescent light source?

Response: The fluorescent and light emitting diode (LED) lamps as the light sources in the experiment. We revised the unit of light intensity from lux to W/m2 throughout the manuscript to avoid confusing with the luminescence light source.

  1. 121. Pentose phosphate pathway is also glycolysis. Do you mean pentose phosphate and Embden-Meyerhof-Parnas pathway?

Response: We changed “glycolysis and the pentose phosphate pathway” to “pentose phosphate and Embden-Meyerhof-Parnas (EMP) pathway” as indicated in (page 4 lines 128-129).

  1. 132, 148, 280, Correct name of C. protothecoides is Auxenochlorella protothecoides.

Response: We revised as recommended (page 4 line 140, page 5 line 157, and page 8 line 283).

  1. 140. Correct name of Scenedesmus dimorphus is Tetradesmus dimorphus.

Response: We revised as recommended (page 4 line 149).

  1. 141, 148, 150, 310, To avoid confusion, please, use the full name for Chlorella and Coelastrella. Because C. could refer to Chlorella, Coelastrella and Chlorococcum.

Response: We revised as recommended (page 4 line 149,page 5 line 156, and page 11 line 336).

  1. 144. Beta should be β.

Response: We revised as recommended (page 4 line 152).

  1. 150. Correct name of Chlorella minutissima is Mychonastes homosphaera.

Response: We revised as recommended (page 5 line 159).

  1. 150. D. salina should be written in full.

Response: We revised as recommended (page 5 line 159).

  1. 150. It was only in the case of luminescent lamps. How can it be explained?

Response: The xylose inhibition was observed in both treatments with fluorescent and LED lamps. We made mistake with the colors of the lines for xylose and arabinose for fluorescent and LED treatments (in Figure 1). This might make the reader to be confused. Therefore, we made a correction as shown in Figure 1.

  1. 155. “phosphorus uptake in photosynthesis” - what is it? Photosynthesis is the process in which a phototrophic cell uses sunlight to synthesize nutrients from carbon dioxide and water.

Response: We revised the sentence to “Xylose or phosphorylated derivatives can slow microalgal growth by competing with xylulose 5-phosphate in the transketolase enzymatic system and blocking the photosynthetic carbon cycle. (page 5 lines 163-165)”. The reference for this statement was added to the text and reference list (page 18 lines 622-624).

  1. 161-162. For this statement emission spectra should be given as supplementary materials.

Response: The detail on emission spectra for fluorescent and LED light sources were presented in the reference that we cited (Schulze et al., 2014) (page 18 lines 627-628). To avoid plagiarism, we designed not to add the spectra from the reference paper to our manuscript. 

  1. Which sugar was selected for further studies as a result of the subsection 2.1? It should be indicated in the end.

Response: The results form the sugar utilization test (subsection 2.1) indicated that fructose, glucose and sucrose were effective organic carbon sources for microalga Coelastrella sp. KKU-P1 cultivation. Since these sugars are typically contained in molasses, the obtained results indicated the potential of molasses utilization by KKU-P1, and the cultivation of KKU-P1 using molasses as a carbon source was examined in further experiments. We had a short discussion on page 4 lines 141-147.

  1. 177. ‘it is considered’ should be ‘it was considered’.

Response: We revised as recommended (page 5 line 186).

  1. 195-296. It is not clear, how do you define growth/productivity of the culture. For example, the phrases like “Further increasing inoculum concentration to 0.2 g/L led to the reduction of biomass concentration” are confusing. In the case inoculum concentration of 0.2 g/L the biomass concentration is also 0.2 g/L, because you inoculated the biomass.

Response: In order to avoid confusing, we changed “biomass concentration” to “biomass production” throughout the manuscript.

  1. 196-175. What do you mean on feed here?

Response: Feed is a substrate (carbon source). We revise the statment “feed to inoculum ratio” to “substrate to inoculum ratio” throughout the manuscript.

  1. Figure 3a. What are the units of F:I? g/g? Please indicate in the figure and at the first mention in the text.

Response: “F:I” was change to “S:I”. We put the unit of S:I ratio (g/g) in Figure 3 and at the first mention (page 6 line 220) as recommended.

  1. 297-310, Figure 7. Please, indicate, it was the volumetric CO2 percentage (v/v).

Response: We indicated the CO2 concentration unit as %(v/v) (page 10 line 331).

  1. Figure 3,4,5,6. Please, indicate in the legend, how is ‘growth’ defined? As culture dry mass at the female stage of the experiment?

Response: We changed the word “growth” in the figure captions to “biomass production” throughout the manuscript.

  1. 239. What about mixotrophic conditions?

Response: We revised the sentence to “Microalgae cells require sufficient nitrogen to support their growth under photoautotrophic, heterotrophic and mixotrophic conditions.” (page 7 line 249 - Page 8 line 262).

  1. 272, 395. It is not cell concentration, but biomass concentration.

Response: We revised “cell concentration” as “biomass production” throughout the manuscript to avoid confusing.

  1. 250-282, 413. Figure 5. lux are units of illuminance, not light intensity. Units of light intensity are W/m2.

Response: We converted units from “lux” to “W/m2” throughout the manuscript.

  1. Table 1. There are some questions about the amino acid profile. First, what is about asparagine, glutamine and selenocysteine? Second, how can you explain the absence of free tryptophan? Coelastrella has many W-containing proteins.

Response: We are very sorry that we made mistake with the name of amino acids we analyzed. We corrected “glutamic acid” to “glutamine” and “aspartic acid” to “aspartate”. However, we did not analyze asparagine and selenocysteine in this study. We gave this statement to the manuscript on page 11 lines 349-350. The discussion regarding tryptophan content in Coelastrella sp. was also provided in page 11 lines 349-356.

  1. 291-296. Please, compare you results on CO2 effect with other works on Coelastrella spp. (doi.org/10.1134/S1021443716040105, doi.org/10.1016/j.bcab.2018.03.022).

Response: We compared the results with the literature and added discussions regarding the effect of CO2 as recommended (page 10 lines 325-328).

  1. 310. Correct name of Chlorella kesslerii is Parachlorella kesslerii. Correct name of Haematococcus pluvialis is Haematococcus lacustris.

Response: We revised as recommended (page 11 line 336).

  1. 342. Please, compare with other data on Coelastrella.

Response: We compared the results with the literature and added discussions regarding the pigment contents of Coelastrella sp. as recommended (pages 11-12 lines 367-373).

  1. 344-352. Please, discuss fatty acid composition compared to other studied Coelastrella spp. (doi.org/10.3390/life12040560, doi.org/10.1134/S1021443716040105, doi.org/10.4490/algae.2017.32.8.6, doi.org/10.1016/j.bcab.2018.03.022, doi.org/10.3390/life12030334).

Response: We compared the results with the literature and added discussions regarding the fatty acid proflies of Coelastrella sp. as recommended (page 12 lines 385-389).

  1. 364-383. The weakness of the methodology is using mass concentrations of sugars instead of molar ones. Obviously, the same mass of pentoses and hexoses contains different amounts of sugar molecules. It could affect growth. This limitation should be mentioned; I suggest recalculate molar concentrations.

Response: We calculated and added the molar concentration of sugars as recommended (page 13 line 412)

  1. 379. μl should be μL.

Response: We revised as recommended (page 13 line 414).

  1. 387-389. Have you studied or have you found published data on composition of used molasse? It would be nice to discuss it in Methods and/or Discussion.

Response: We gave short discussion statement regarding molasses compositions as shown in page 14 Lines 430-431

  1. 427. %-v/v?

Response: It is % v/v. We revised as recommended (page 15 line 468).

Round 2

Reviewer 2 Report

The Authors have signifficantly improved the manuscript and have satisfactory answered most of my comments. I have retained two original issues about culture productivity and amino acid composition.  There is also minor commeny on the newly inserted text.

1. Please defind in the Abstract and the main body of the text at the first mantion the term 'culture productivity'. What is it? In biotechnology, culture productivity implies the presence of units of time, whereas in work it has the dimension of g/L. For what period of time was such biomass accumulated? The definition should address this issue.

2. Revise names of amino acids and presentation of this data. I understand the responses about triptophane and aparagine. But it is not the case of glutamic acid and glutamin. In case, you actually studied glutamine, what is about glutamic acid? Aspartate and aspartic acid are im principle, the same, because the first is a salt of the second, but for uniformity with others, the name aspartic acid would be more appropriate.

3. Do you actually mean Coelastrella spp. (not Coelastrella sp.), because these references are aboult different Coelastrella species (l. 33-44)?

After fixing these minor issues the article can be accepted.

Author Response

  1. Please defind in the Abstract and the main body of the text at the first mantion the term 'culture productivity'. What is it? In biotechnology, culture productivity implies the presence of units of time, whereas in work it has the dimension of g/L. For what period of time was such biomass accumulated? The definition should address this issue.

Response : We did not mention the term culture productivity in the manuscript. However, the biomass production results provided in subsection 2.2.2 – 2.2.4 and section 2.3 were obtained at the end of cultivation noting that the cultivation was conducted until reaching the stationary phase. We provide this information in the manuscript as indicated lines 208, 223, 247 and 442.

  1. Revise names of amino acids and presentation of this data. I understand the responses about triptophane and aparagine. But it is not the case of glutamic acid and glutamin. In case, you actually studied glutamine, what is about glutamic acid? Aspartate and aspartic acid are im principle, the same, because the first is a salt of the second, but for uniformity with others, the name aspartic acid would be more appropriate.

Response : We apologize for this mistake again.  The glutamic acid was actually analyzed in our experiment, therefore, we made correction as indicated in line 351 and Table 1. Aspartate was changed to aspartic acid as recommended (line 351 and Table 1).

  1. Do you actually mean Coelastrella (not Coelastrellasp.), because these references are aboult different Coelastrella species (l. 33-44)?

Response : We actually mean Coelastrella spp. and the term “sp.” was revised to “spp.” when various Coelastrella sp. was mentioned in the manuscript as indicated in lines 33, 35, 37, 40, 329, 376, 389 and 391.